# Respiro: Continuous Respiratory Rate Monitoring During Motion via Wearable Ultra-Wideband Radar

Sebastian Reidy*, Manuel Meier*, Christian Holz

*Abstract*—Deviations in respiratory rate often precede abnormalities in other vital signs. However, continuously monitoring respiratory rates outside clinical settings remains challenging due to the obtrusive nature and sensitivity to body motions in existing monitoring approaches. In this study, we propose a single-point-of-contact wearable device that leverages off-the-shelf, consumer-grade ultra-wideband radar to monitor respiratory rate as part of a chest strap. Our signal processing pipeline reliably extracts the wearer's respiratory signal from windowed complex channel impulse responses. In a controlled experiment, twelve participants performed various activities to evaluate the system's accuracy under motion while capturing ground-truth recordings through a spirometer. Our method extracted respiratory rates with less than 1 breath per minute deviation in 71% of all measurements, averaging 1.11 breaths per minute across all sessions and participants. Our findings underscore the potential of consumer-grade ultra-wideband radar technology in body-worn devices for unobtrusive yet effective respiratory monitoring.

*Index Terms*—breath monitoring, breath rate, respiration monitoring, respiration rate, single point of contact, UWB, wearable ultra wideband radar

## I. INTRODUCTION

RESPIRATORY rate (RR), along with heart rate, blood pressure, and body temperature, is a primary vital sign routinely assessed by medical professionals. RR measurements are essential in diagnosing conditions such as acidosis or pneumonia [1], [2]. Additionally, an elevated RR over the course of 24-72 hours can predict severe adverse events, including cardiac arrest and the need for intensive care unit admission [3], [4]. In everyday contexts, RR can provide insights into stressors like emotional load, heat, or physical effort [1]. Traditionally, capnometry and spirometry are considered the gold standards for respiration monitoring. However, these methods require air masks, limiting their suitability for long-term monitoring [5], [6]. Alternative methods, such as respiration belts and the use of inertial measurement units (IMUs), have emerged, but they are less accurate than spirometry or capnography and are prone to motion artifacts.

Amidst these challenges, ultra-wideband (UWB) radar has garnered interest over the past two decades since the Federal Communications Commission (FCC) opened the 3.1 to 10.6 GHz spectrum for medical imaging [7]. There is substantial research on non-contact RR recovery using UWB radar [8]–[10], including through debris [11] or walls [12]. This remote monitoring detects chest wall motion using UWB beacons in a person's environment. However, these methods are prone to

*These authors contributed equally to this work. All authors are affiliated with the Department of Computer Science, ETH Zurich, Zürich, Switzerland. E-mails: {firstname.lastname@inf.ethz.ch, e.g. christian.holz@inf.ethz.ch}

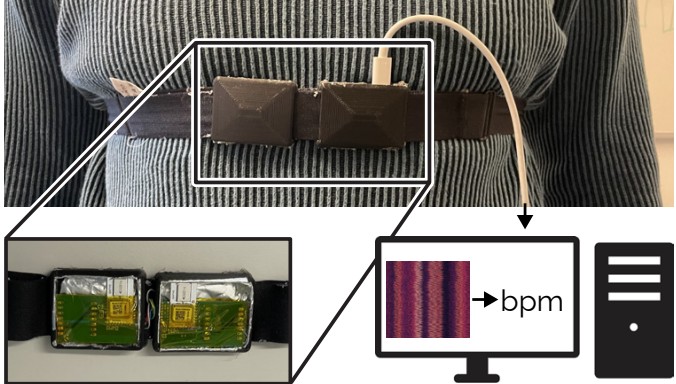

Fig. 1. Our wearable device placed on the sternum uses an off-the-shelf ultra-wideband radar module to produce reliable measurements of the respiratory rate.

errors due to unrelated body motion and are unsuitable for tracking the RR during activities and in changing environments. UWB radar can also probe the body itself, detecting varying dielectric properties of tissues which affect UWB pulses differently [13]. Specifically, this applies to the lungs, which exhibit different dielectric properties when inflated versus deflated: more signal is reflected at the muscle-lung interface when the lung is inflated compared to its deflated state [14]. This is supported by Cavagnaro et al. who studied the propagation of UWB pulses into human tissue and proposed that effective RR recovery can be achieved by capturing the air-skin reflection signal with an antenna positioned 1 meter away from the body. They suggested that on-body antennas could potentially recover the RR from a small signal reflected by the posterior lung wall [15]. However, on-body UWB radar for RR monitoring has only been studied using laboratory equipment [16], [17], using two points of contact [18], [19], or with custom UWB antennas integrated into a seat [20], all of which were evaluated exclusively on stationary users.

Here, we contribute Respiro, the first single-point-of-contact wearable device for RR measurement as shown in Figure 1, using consumer-grade, off-the-shelf UWB radar modules. We also present an offline data processing pipeline that computes RR from in-body reflections contained in complex channel impulse response (CIR) estimates. We evaluated our system in a user study where participants engaged in various activities to induce motion artifacts. This approach aligns with other studies assessing wearable devices for respiratory monitoring [21], [22]. To the best of our knowledge, this is the

first study to evaluate a wearable UWB-based RR detection system on moving participants. Additionally, we collected accelerometer data and compare our UWB-based system with the accelerometer-based systems of Bates et al. [23] and Rahman et al. [24].

### A. Related Work

Reviews on RR monitoring techniques differentiate between contact and non-contact methods [5], [25]. We adhere to this categorization and provide an overview of respiratory monitoring techniques and subsequently focus on related work leveraging UWB radar for RR monitoring.

*1) Non-contact-based RR monitoring:* Methods have been published using a variety of sensing modalities. Wang et al. utilized ultrasound to sense exhaled airflow in sleeping subjects, achieving a median error of less than 0.3 breaths per minute (bpm) [26]. Focussing on recovering the RR from chest movements, Bernacchia et al. used the Microsoft Kinect system, which combines a depth sensor, video camera, and microphones, reporting a standard deviation of the residual RR of 9.7% [27]. Radar technology has also been employed to observe chest movements induced by respiration. Researchers have developed various system architectures using frequency-modulated continuous-wave (FMCW) radar at radio and laser frequencies, where respiration-induced chest movements alter the phase and frequency of the carrier signal [5].

*2) Non-contact-based RR monitoring with UWB radar:* Wang et al. conducted a comparative analysis between FMCW radar and UWB radar for non-contact vital sign extraction in varying conditions regarding distance and orientation of the subject as well as obstructing obstacles. Their findings indicated that UWB radar offers higher accuracy and a better signal-to-noise ratio (SNR) than FMCW radar, showcasing the potential of UWB radar for RR monitoring [10]. Immoreev et al. demonstrated the ability to recover RR from tiny chest movements using a UWB radar mounted 2 meters above a hospital bed [9]. In more complex environments, such as disaster sites, Li et al. utilized the curvelet transform to remove antenna coupling and background clutter, followed by singular value decomposition to denoise the respiratory signal before extracting RR with a fast Fourier transform [11]. Duan et al. advanced this approach with an algorithm based on variational mode decomposition, capable of recovering RR in open spaces and through walls, achieving a correct detection rate of 95% [28]. Regev et al. further improved accuracy by fusing UWB radar signals with data from an RGB camera, resulting in a maximum error of 0.5 bpm when testing on 14 sitting participants [29]. Li et al. explored another method by using two radar modules (DWM1000, Qorvo) to recover RR from sitting subjects at different positions in a room. They investigated two approaches for computing the respiration signal from reflections at different spatial distances and concluded that a linear combination of reflections from multiple spatial distances outperformed selecting reflections solely from the most probable spatial distance [30]. This linear combination approach leveraged the advantages of capturing comprehensive

respiratory patterns, thereby enhancing the robustness and accuracy of RR monitoring in varied environments.

Non-contact systems are typically constrained to stationary settings indoors. Our method, by contrast, is designed for real-world scenarios where individuals may be engaged in physical activities and not constrained to a previously equipped environment.

*3) Contact-based RR monitoring:* These methods for RR monitoring encompass an even broader range of sensing modalities compared to non-contact methods. Kumar et al. utilized wearable microphones to estimate RR [31]. Their setup, tested on subjects before, during, and after exercise, achieved a mean squared error in the RR of 0.2 breaths per minute (bpm). Basra et al. developed a system that recovered RR using a temperature sensor on a nose clip, exploiting the temperature difference between inhaled and exhaled air [32]. Similarly, Guder et al. monitored RR by sensing humidity differences in an air mask [33].

Further exploring physiological signal variations, Heydari et al. derived RR from changes in bio-impedance measured on electrodes placed on the shoulder of participants, reporting an error of $\leq 1$ bpm across different breathing patterns (slow, fast, deep, hold, and normal) [34]. Another approach leverages respiratory sinus arrhythmia, where the heart rate increases during inhalation and decreases during exhalation. This phenomenon enables RR measurement through heart rate monitoring techniques such as electrocardiography (ECG), photoplethysmography (PPG), ballistocardiography, and seismocardiography [6]. Charlton et al. evaluated 270 algorithms using ECG and PPG signals, finding that the best-performing algorithm had a 95% limit of agreement of -4.7 to 4.7 bpm [35].

Moving to inertial sensor-based methods, accelerometers, gyroscopes, and respiration belts can detect periodic thorax volume changes during breathing [6]. Rahman et al. used a single accelerometer mounted on a belt on the thorax to estimate RR, achieving successful recovery in 97% of attempts across five respiratory rates on stationary subjects. However, this approach did not address motion artifacts, a known limitation of accelerometer-based systems [21], [24]. To mitigate motion artifacts, Bates et al. reconstructed RR from the angular velocity of the current rotation angle measured relative to the mean direction of gravity. They implemented a movement detection method to pause their system during motion, only computing RR when the subject was at rest [23].

*4) Contact-based RR monitoring with UWB radar:* Pitella et al. explored single-point-of-contact antennas placed on the chest to capture individual UWB pulses reflected by the human body, concluding that the reflected signal changes with the respiration phase [16]. Building on this, they investigated various signal processing techniques for extracting RR from in-body reflected pulses, finding that mean subtraction offered favorable accuracy and fast processing [17]. Notably, their UWB radar setup could detect RR even during arm movement, unlike a piezoelectric belt which picked up the arm movement rate. However, their setup involved a vector network analyzer, making it non-wearable. Schires et al. optimized UWB an-

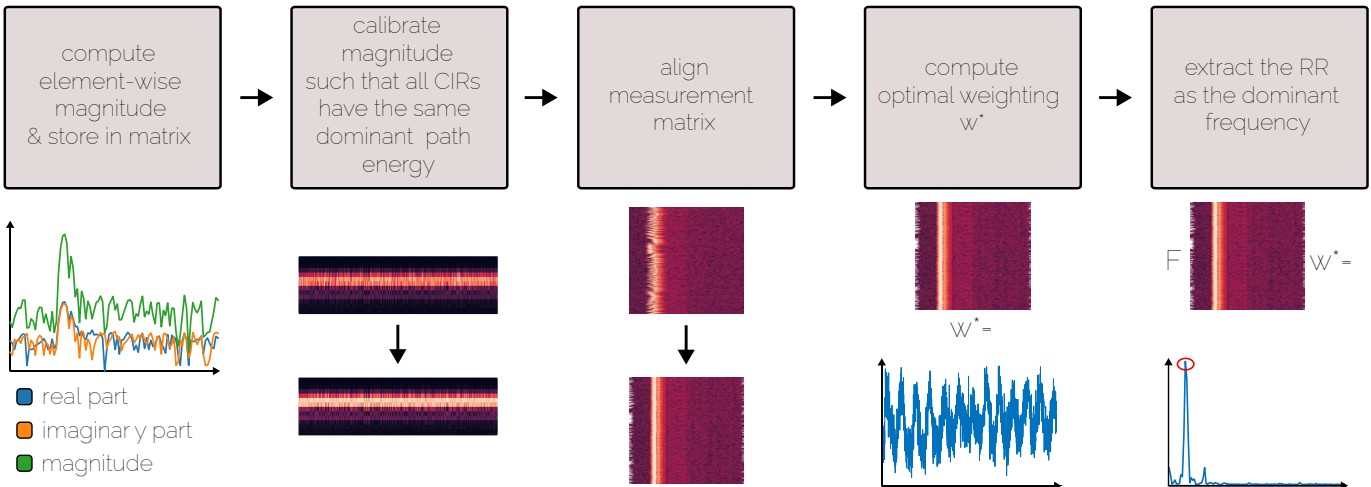

Fig. 2. The processing pipeline includes (left to right): Computation of the magnitude of the complex Channel Impulse Response (CIR) and storage as a row in a matrix, then the CIRs are calibrated, and aligned. Finally, an optimal weighting to maximize the energy in the relevant frequency bands is computed, applied and the dominant frequency computed.

tennas to maximize power transmission into the human body by embedding the antennas in the backrest of a car seat, recovering RR from phase variations in the reflected pulse [20]. Culjak et al. introduced a two-point-of-contact wearable RR monitoring system, positioning one radar module (DWM1000, Qorvo) on the front and another on the back of the thorax [18], using the CIR as input similar to Li et al. [30]. They employed mean subtraction to eliminate static components and applied a bandpass filter to the band of possible respiration frequencies in slow time. For each slow-time CIR, they identified the first four extrema in fast time and computed the difference in slow time between those fast time indexes, finally using an FFT to identify the dominant frequency. They reported a relative error of 10% of the RR in the optimal configuration [18]. They later achieved a root mean squared error below 0.2 bpm using two data fusion algorithms—naive Bayes inference and Kalman filtering—to fuse UWB signals with accelerometer data [19].

While some contact-based methods achieve high accuracy, they often require obtrusive sensors, and less obtrusive IMU-based methods struggle with motion artifacts. UWB radar has recently gained attention as a viable alternative, offering accurate RR detection even in the presence of movement. However, current UWB radar based systems are not wearable or require multiple points-of-contact. To overcome these limitations, we have developed a novel wearable system utilizing single-point-of-contact UWB radar technology.

## II. METHOD

### A. Hardware

Respiro includes two UWB radar modules (DWM3000, Qorvo), the same series as the one integrated into commercial devices such as the Google Pixel 6 Pro. Due to their half-duplex nature, two modules are required as each one cannot transmit and receive simultaneously. The radar modules are controlled by a microcontroller (ESP32-C3, Espressif Systems) through the serial peripheral interface (SPI) protocol,

configuring them for transmitting and receiving states and acquiring measurement data. This data is then transmitted by the microcontroller to a computer for further processing via serial communication. Each radar module is soldered onto a separate PCB. The two PCBs are enclosed in 3D-printed cases lined with aluminum foil on all surfaces that do not face the wearer, reducing the direct path component of the signal in comparison to in-body reflections.

### B. Sampling

Rather than raw reflected pulses, the radar module only provides access to a complex CIR estimate which can be used for RR measurement purposes. The CIR, represented as $h[m]$, elucidates the relationship between a probing signal $x[m]$ and its reflection $y[m]$:

$$y[m] = \sum_{k=0}^{M} h[k]x[m-k] \tag{1}$$

Physically, $h[m]$ illustrates the reflection as if the probing pulse were a Dirac delta function $x[m] = \delta[m]$. $M$ denotes the number of samples in a CIR.

Each transmitted packet begins with an Ipatov preamble, followed by a start-of-frame delimiter. The packet body may include a physical header, followed by a payload, a scrambled timestamp sequence, or both. The radar module estimates a CIR for every packet by exploiting the perfect autocorrelation property of the Ipatov preamble. We transmit a counter value during our measurements to avoid duplicating CIR samples. The CIR preamble spans 1016 complex samples when using the radar module's default pulse repetition frequency of 64 MHz. The sampling period of the CIR equates to half the fundamental period $T_{CIR} = (2 \cdot 499.2\,\text{MHz})^{-1} \approx 1\text{ns}$.

A UWB pulse propagates through the air at a velocity of $v_{air} = \frac{c}{n_{air}} = 29.9\frac{\text{cm}}{\text{ns}}$, where $c$ is the speed of light in a vacuum and $n_{air} \approx 1$ the refractive index of air.

Various tissues within the human body have distinct dielectric properties. Prior work has used an average refractive index of $n_{thorax} = \sqrt{50}$ to model the thorax [15]–[17]. Under this assumption, the pulse propagation speed within the thorax is $v_{thorax} = \frac{c}{n_{thorax}} = 4.2 \frac{cm}{ns}$. Consequently, each sample corresponds to a spatial distance of $4.2$ cm within the human body. We extracted 100 samples, aligning with previous work where 300 and 75 samples after the direct path were used ([18], [30]). However, the direct path component commences not at index 0 of the extracted CIR: In a random sample of more than 26 000 CIR samples recorded over 14 minutes, we discovered that the average index of the direct path, i.e., the index in the CIR with the highest magnitude, is approximately 741.7 with a standard deviation of 2.4. The lowest observed direct path index was 735, and the highest was 746. Therefore, we sampled 100 indices from the CIR starting at an offset of 720. This sampling strategy ensures over 70 samples to include reflections from within the thorax, corresponding to thorax diameters up to $\frac{70}{2} \cdot 4.2 \frac{cm}{sample} = 148.3$ cm. This reduces memory and communication demands by over 90% compared to sampling the entire CIR while retaining pertinent information.

### C. Data Processing

Figure 2 provides an overview of the data processing pipeline of Respiro. Initially, we compute the element-wise magnitude of the complex-valued CIR and store a predefined time window as a row in a measurement matrix $\mathbf{H}$. The rows of matrix $\mathbf{H}$ represent estimated reflections from a Dirac pulse at different temporal instances, while the columns correspond to reflections originating from distinct spatial distances.

Since not all direct path components possess the same energy, we adopted the dominant path approach proposed by Li et al., where they calibrated the CIR of a WIFI system for extracting RRs [36]. Their method involves computing a calibration coefficient based on the energy of the direct path component, which they extracted from a window around the time domain peak of the CIR. In our configuration, we utilize a 7 ns window to capture the energy of the directed path. The calibrated CIR $h'[m]$ is derived from the CIR $h[m]$ as follows:

$$h'[m] = \left( \frac{1}{2D+1} \sqrt{\sum_{|\tau - \tau^*| \leq D} |h[\tau]|^2} \right)^{-1} h[m] \quad (2)$$

Where $D = 3$ is the number of indices in the window left and right of the index of the CIR peak $\tau^* = \underset{\tau}{\operatorname{argmax}} |h[\tau]|$.

To address the issue that the extracted CIRs do not contain their direct path component at the same index every time and the rows of $\mathbf{H}$ are thus not perfectly aligned, we employ cross-correlation maximization to align the rows of $\mathbf{H}$. The initial CIR $h_0[m]$ in $\mathbf{H}$ serves as a reference and each subsequent CIR $h_n[m]$ is shifted by an index $k_n$ to achieve maximum cross-correlation.

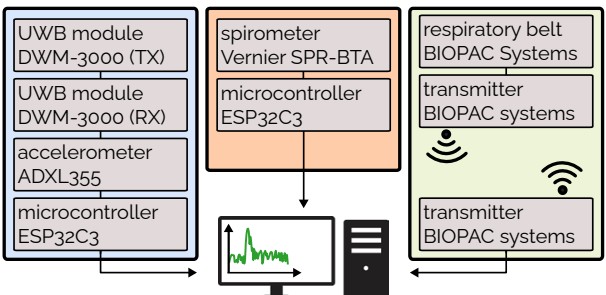

Fig. 3. The data capturing system for the user study included the custom UWB device (left), a spirometer ground truth (center), and a respiratory belt (right).

$$k_n = \underset{k}{\operatorname{argmax}} \left| \sum_m^M h_0[m] \, h_n[m+k] \right| \quad (3)$$

Various anatomical structures within the human body, such as the anterior and posterior lung walls, act as potential sources for reflections containing RR information. We extract the RR from a linear combination of reflections across all spatial distances, denoted as $\mathbf{H}w$. The weight vector is computed using an optimization criterion introduced by Li et al. [30] when they successfully extracted respiration signals from individuals at unknown positions within a room. They maximized the energy within the band of possible respiration frequencies while maintaining the overall energy constant by computing the optimal weighting $w_{opt}$ as follows:

$$w_{opt} = \underset{w}{\operatorname{argmax}} \, (\mathbf{F_I H}w)^H (\mathbf{F_I H}w) \quad (4)$$

$$\text{s.t. const.} = (\mathbf{FH}w)^H (\mathbf{FH}w) \quad (5)$$

where $\mathbf{F}$ is the discrete Fourier transform (DFT) matrix and $\mathbf{F_I}$ is the DFT matrix containing frequencies in the band of possible respiration frequencies. This leads to a dynamic selection of the measurement depths in the body where respiratory signals occur. We used a band of possible respiration frequencies of $f_r^l = 0.1$ Hz (6 bpm) to $f_r^h = 0.7$ Hz (42 bpm) which enables the detection of respiratory rates both at rest and during physical exercise. Ultimately, we computed the Fourier transform of $\mathbf{H}w_{opt}$ and identified the respiration frequency as the frequency exhibiting the highest energy density within the band of possible respiration frequencies.

### III. CONTROLLED EVALUATION

#### A. System Overview

As shown in Figure 3, the study setup included the custom UWB RR detection device, a spirometer (SPR-BTA, Vernier), a respiration belt (BN-RESP-XDCR, BIOPAC Systems) with peripherals, and a computer for data collection and processing. We integrated an accelerometer (ADXL355, Analog Devices) with the UWB device to enable comparison with accelerometer-based systems proposed by Bates et al. and Rahman et al. [23], [24]. The on-board microcontroller

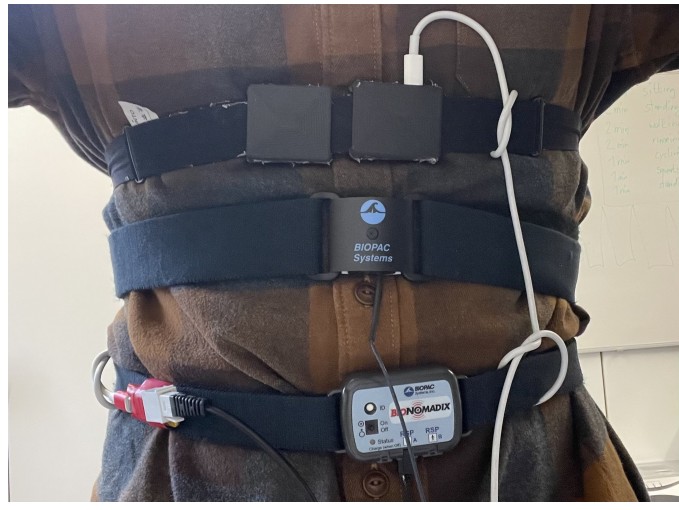

Fig. 4. The device setup on the participant's thorax for the study included the custom UWB device which was strapped over the sternum, followed by the reference respiratory belt over the abdomen, and the transmitter of the respiratory belt.

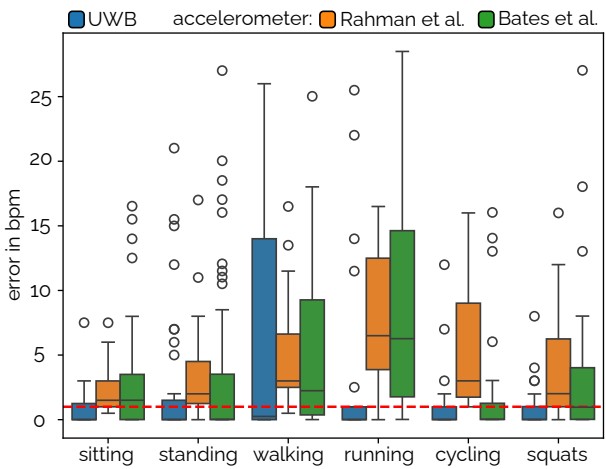

Fig. 5. The error distributions of our UWB-based approach to detect respiratory rate compared to the accelerometer-based approaches of Rahman et al. [24] and Bates et al. [23]. The red dashed line indicates the error threshold of 1 bpm used to classify successful traces.

connects to the accelerometer and the UWB radar modules and facilitates data transmission to the computer via USB-C. Similarly, the microcontroller associated with the spirometer samples the analog signal and transmits the acquired data to the computer via USB-C. The respiration belt system features an independent transmitter for wireless data transmission to the base station, which subsequently transmits the data to the computer through WIFI.

As shown in Figure 4, we positioned the UWB device on the participants' sternum to ensure proper placement over the lungs while minimizing interference with upper body movement. Following the manufacturer's guidelines, we situated the respiration belt over the abdomen. To prevent contact during movement and mitigate data artifacts, we maintained a 1–2 cm gap between the respiration belt and the UWB device. The respiration belt transmitter was placed with a sufficient gap to the respiration belt to avoid contact during participant movement. To prevent accidental sensor detachment during exercise, we secured the USB-C cable of the UWB device to both the supporting belt of the device and the belt of the respiration belt transmitter. Similarly, we affixed the microcontroller interfacing with the spirometer to the transmitter belt and secured its USB-C cable to the transmitter belt.

In the user study, we acquired CIR and accelerometer data at a sampling frequency of 32 Hz, following prior work which acquired data at rates ranging from 19.3 Hz to 65 Hz [18], [20], [30]. The analog signal from the spirometer was sampled at 50 Hz using the built-in analog-to-digital converter on the connected microcontroller. Additionally, the respiration belt signal was acquired at its default frequency of 2 kHz.

### B. Participants

The participants were 12 healthy adults, recruited on a voluntary basis. 3 female and 9 male participants took part, with ages ranging between 21 and 29 (mean age: 25).

### C. Procedure

Upon arrival, participants were introduced to the experimental protocol. An experimenter outfitted them with the recording devices. The participants then tested the treadmill and the experimenter adjusted the walking and running speeds if necessary, starting at 5 km/h and 10 km/h respectively. For the data acquisition, the participants then completed the following activity sequence twice:

1) two minutes of sitting on a chair
2) two minutes of standing still
3) two minutes of walking on a treadmill
4) two minutes of running on a treadmill
5) one minute break (or more at participant's request)
6) one minute of cycling on an exercise bike
7) one minute break (or more at participant's request)
8) one minute of squats
9) one minute of standing still

For the second iteration, the spirometer was removed.

The experimental procedure was reviewed and approved by the ETH Zurich Ethics Commission under the application reference EK-2023-N-183.

## IV. RESULTS

For analysis, we considered the recordings of the seven activities described in the study procedure, excluding breaks. With 12 participants each performing two iterations of the protocol, this resulted in $12 \cdot 14 = 168$ traces. Two traces were excluded from analysis: one due to a participant unintentionally breaching the study procedure during the sitting sequence, and another due to a respiration belt failure during the standing sequence.

Table I shows the aggregated results of the user study categorized by activity. We report the SNR, the RMSE, the mean absolute percentage error (MAPE), and the share of traces exhibiting an error of less than 1 bpm. We compute the

| | UWB SNR | UWB MAPE | UWB RMSE | UWB error <1 bpm | Rahman et. al [24] RMSE | [24] error <1 bpm | Bates et al. [23] RMSE | [23] error <1 bpm |
|---|---|---|---|---|---|---|---|---|
| **w/ spirometer as reference** | | | | | | | | |
| 120 s sitting | 3.09 | 3.96 | **1.08** | **0.82** | 3.18 | 0.36 | 5.28 | 0.73 |
| 120 s standing | 1.84 | 6.83 | **2.16** | 0.67 | 3.63 | 0.33 | 6.31 | **0.75** |
| 120 s walking | 1.08 | 6.47 | **4.49** | 0.83 | 7.39 | 0.17 | 9.49 | 0.42 |
| 120 s running | 2.10 | 0.43 | **0.32** | 0.92 | 8.84 | 0.08 | 11.00 | 0.33 |
| 60 s cycling | 2.26 | 5.08 | 2.29 | 0.75 | 5.04 | 0.17 | **0.65** | **0.83** |
| 60 s squats | 1.60 | 4.51 | **2.50** | 0.75 | 3.79 | 0.33 | 6.24 | 0.50 |
| 60 s standing | 1.31 | 13.63 | 6.24 | 0.83 | **3.95** | 0.17 | 6.05 | 0.67 |
| **w/o spirometer, respiration belt as reference** | | | | | | | | |
| 120 s sitting | 1.75 | 6.45 | **2.47** | **0.67** | 2.80 | 0.42 | 7.48 | 0.08 |
| 120 s standing | 0.80 | 15.76 | 6.00 | 0.45 | **3.39** | 0.09 | 9.26 | **0.64** |
| 120 s walking | 0.49 | 36.57 | 15.29 | **0.50** | **5.20** | 0.17 | 9.20 | 0.25 |
| 120 s running | 0.81 | 29.34 | 11.07 | **0.58** | **9.25** | 0.08 | 11.87 | 0.08 |
| 60 s cycling | 1.16 | 4.36 | **3.52** | **0.83** | 8.15 | 0.25 | 7.48 | 0.50 |
| 60 s squats | 1.13 | 4.54 | **1.61** | 0.67 | 6.99 | 0.33 | 8.72 | **0.67** |
| 60 s standing | 1.65 | 10.19 | **5.13** | **0.67** | 6.76 | 0.42 | 8.21 | **0.67** |

TABLE I

AGGREGATED USER STUDY RESULTS, COMPARING OUR METHOD TO THE ACCELEROMETER-BASED METHODS OF RAHMAN ET. AL [24] AND BATES ET AL. [23]. *RMSE* IS REPORTED IN BPM, *MAPE* AS A PERCENTAGE, *error ¡ 1bpm* AS A SHARE

SNR of the processed UWB signal, the respiration belt signal, and the spirometer signal using the methodology outlined by Droitcour et al. [37], wherein signal energy is determined as the energy density within a range of six bpm around the dominant frequency, while the remaining energy is considered noise. We report the RMSE and the count of successful traces, i.e., those with an error lower than one bpm, consistent with Culjak et al. and Rahman et al. [18], [24]. Furthermore, Table I also incorporates results from the accelerometer-based pipelines from Rahman et al. and Bates et al. [23].

The processed UWB signal generally exhibits an SNR greater than 1, except during the standing, walking, and running activities in the second round of the study. Both the spirometer and respiration belt ground truth signals consistently demonstrate an SNR greater than 1 when averaged across participants. Success rates range between 0.45 and 0.92 across activities which is better than the accelerometer-based approach by Rahman et al. across all activities and at least equal to Bates et al.'s approach in 11 of 14 cases investigated.

Figure 5 illustrates box plots representing the distributions of absolute errors for our UWB-based approach and the accelerometer-based approaches. Across all activities, the median error of our UWB-based system remains below the 1 bpm threshold, and the error distribution is narrower than the ones of the accelerometer-based approaches of Bates et al. and Rahman et al. For walking traces, our UWB system demonstrates a more scattered distribution of absolute errors compared to Bates et al. and Rahman et al.'s systems.

### A. Effect of the Spirometer

The study protocol consisted of two identical activity sequences, where a spirometer was used only during the first iteration. Everything else remained the same including the activities and sensor positioning.

To compare the results of both settings, we used the respiration belt signal from both rounds as the ground truth for this analysis. When wearing a spirometer, the mean absolute error (MAE) decreased by 2.65 bpm (3.76 bpm to 1.11 bpm, median absolute error 0.016 to 0.006), and the success rate showed an increase of 0.20. Additionally, the SNR of the UWB method increased by 70% on average. Computing the same SNR for the respiratory belt reference signal results in an increase of 98% when additionally using a spirometer compared to the procedure iteration without.

### B. Accuracy of the respiration belt

During the first round of the user study, we captured signals from the spirometer and the respiration belt. Comparing the respiration belt to the spirometer, it achieves a success rate ($<$ 1 bpm error) of 0.84% overall with values ranging from 0.67 (squats) to 1.00 (running). The RMSE across activities is 3.06, inflated by the walking and squats activities with RMSE of 4.54, and 6.40 respectively.

### C. Correlation of Success and SNR

For the UWB-based method, we observe a negative correlation between the SNR and the absolute error with a correlation coefficient of -0.29. This means that traces with a high SNR in the UWB signal tend to have a lower error. Similarly, when discriminating traces at SNR = 1, the mean success rate is 0.86 among traces with SNR $>$ 1 and 0.56 for all others with sample sizes of 84, and 82 respectively.

### D. Effect of the Window Size

We utilized the complete duration of a trace, which spans either one or two minutes, for our analyses. Figure 6 depicts how the success rate evolves if we consider windows of shorter duration. We assessed all two-minute traces using sliding window sizes ranging from 2 to 119 seconds (3-second increment), and a step size of 10% of the window size. Across all activities assessed, a trend toward a higher success rate with longer evaluation windows is observable with a steep increase up to a window length of approximately 35 seconds and a flattening curve afterward. The breadth of the confidence

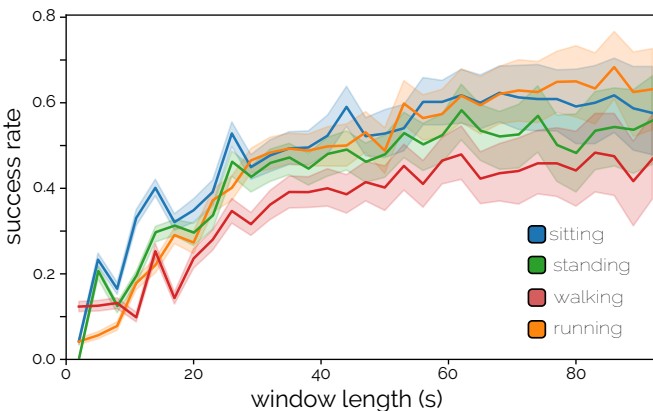

Fig. 6. The success rate, i.e., error ≤ 1 bpm (95% confidence interval), depending on the window size.

interval expands with longer window sizes where the number of available windows for analysis decreases.

## V. DISCUSSION

We presented Respiro, a single-point-of-contact, wearable UWB radar system to monitor the RR. The method worked not only in static settings but also during activities where it was less prone to motion artifacts compared to methods based on accelerometers. Using UWB modules that are used in commercial devices such as smartphones, we were able to measure the respiratory rate not only in static settings but also during physical activities. By not requiring lab equipment or complex custom-built circuits and antennas, we show the potential of this sensing modality to be used in unobtrusive wearable devices monitoring the RR. The most closely related prior work by Culjak et al. reported a mean absolute percentage error (MAPE) of 8.75% when analyzing a transmissive UWB signal through the chest while the participants were sitting in a chair. In the same setting in our study (sitting participant), we observed a MAPE of 6.45%, highlighting the viability of a wearable single-point-of-contact UWB radar system in monitoring the RR, achieving accuracy comparable to that of a wearable two-point-of-contact system. Compared to the accelerometer-based systems of Rahman et al. and Bates et al., our UWB-based approach has a higher success rate by being less prone to motion artifacts. However, movement of the sensor encasement can still lead to motion artifacts in the measurements. This is particularly evident during walking activities, where the cadence range overlaps with possible respiratory rates. Limitations of our study include the relatively small sample size of 12 participants and the limited age diversity. Future research should include a larger number of participants to investigate participant-specific effects, such as variations in body types, on the results.

### A. Ground Truth with Spirometer and Respiration Belt

The acquisition of viable ground truth respiratory rate measurements is challenging. The use of a spirometer introduces resistance to the airflow during breathing and leads to the inhalation of previously exhaled air still in the mask and the device. These effects potentially prompt subjects to engage in deeper breaths, augmenting the outcomes. We observed this effect in our study where the SNR of the respiratory belt was higher when a spirometer was used at the same time indicating deeper breathing patterns. However, the alternative use of a respiration belt for ground truth measurements is more prone to noise due to motion artifacts or movement of the belt. In our study, when comparing the RR from the respiration belt to the results from the spirometer for reference, the respiration belt yielded the correct respiratory rate in 84.3% of all traces (using 1 bpm tolerance). Both these effects lead to a lower accuracy of the RR detection in the sequence where no spirometer was used. Notably, outliers were more severe when not using the spirometer which explains the substantial difference in the reported MAE. Similar observations have been made in prior work by Pittella et al. who observed that a piezoelectric belt measured arm movements rather than respiration frequencies [17]. This underlines the importance of a well-planned ground truth acquisition in future work toward wearable, unobtrusive RR monitoring devices for everyday use.

### B. Correlation of Success and SNR

Our processing pipeline recovers the respiration signal from which we compute the RR. The SNR estimate from Droitcour et al. serves as a metric to assess the quality of the recovered respiration signal [37]. Consequently, it is reasonable that traces with a higher SNR in the processed UWB signal correspond to a greater success rate in RR recovery. The SNR could thus be used as a predictive measure to estimate the likelihood of successful RR computation before extracting the RR from the processed UWB signal. By incorporating this into the pipeline, the system's performance could be optimized by selectively utilizing computed RR values only when the SNR exceeds a predefined threshold.

### C. Effect of the Window Size

Figure 6 illustrates a higher success rate in RR detection for longer windows of recording data across all activities. This observation is consistent with our expectations, as the optimization problem benefits from a wider reference band. Beyond a window size of 45 seconds, no substantial gains can be observed which may be due to changes in respiratory rate throughout longer window sizes leading to ambiguities.

### D. Limitations

All study participants were aged between 21 and 29 years. While our method adapts to different body types by targeting respiration-related frequency bands, it will be important to include people of a greater age diversity in future efforts. Additionally, our evaluation used only IMU-based baseline algorithms, as most contact-based methods require extra hardware, like custom antennas, which could introduce confounding variables.

## VI. Conclusion

In this work, we presented Respiro, a method to monitor RR using off-the-shelf UWB radar modules as commonly found in devices such as smartphones. Respiro was tested with a custom-built prototype in a study on 12 healthy adults who engaged in multiple activities including walking, running, and cycling which typically produces motion artifacts. When compared with ground truth reference measurements, our method performs on par and sometimes better than more complex measuring methods while being less prone to motion artifacts than accelerometer-based approaches. This showcases the potential for unobtrusive wearable devices that monitor RR in everyday life using UWB radar.

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
