# OpenReview forum: "Respiro: Continuous Respiratory Rate Monitoring During Motion via Wearable Ultra-Wideband Radar"
_IEEE.org/EMBS/BHI/2024/Conference — IEEE BHI'24_

### Official Review · Reviewer_s8hR · 2024-07-29
**Paper review for Respiro: Continuous Respiratory Rate Monitoring During Motion via Wearable Ultra-Wideband Radar**

**Overall Rating:** 6
**Confidence:** 3

**Other Quality Metrics:**

- Clarity of writing: **great**
- Clinical Significance: **good**
- Methodological Novelty: **good**
- Experiments and Results: **great**

**Questions For The Authors:**

- In Fig 4, the device seemed to be worn very tight on the participant's thorax. In future work, it would be better to generate a more precise estimation strategy that could be personalized to patient demographics.

**Strengths:**

- The manuscript provided clear visuals and descriptions of its device setup and framework design.
- The manuscript provided an in-depth overview of the related work
- The manuscript presented significant improvements over baselines on all activities except walking.

**Summary Of The Paper:**

Anomalies in respiratory rate could indicate possible abnormalities in other vital signs, but monitoring RR in out-of-clinical settings is bottlenecked by bulky, specialized devices and user comforts. The manuscript proposed Respiro as a continuous remote RR monitoring framework using only single-point-of-contact consumer-grade wearable sensors. It evaluated Respiro's accuracy and reliability under various activities via a controlled study and reported on-par performance against previous, complex approaches.

**Weaknesses:**

- In Section I.A., while the literature review on related work was very in-depth, the manuscript didn't explain the limitations and improvements between different categorized approaches. I recommend authors conclude each related work category with its weakness and connect it to their motivations/innovations in designing this framework.
- Demographics such as Age and BMI are important factors that impede the framework's reliability of RR estimation. However, the manuscript's study cohort was very homogenous, with only healthy young adults under 30 years old. It would be great to see future experiments addressing these limitations.
- In Section II.B., the manuscript could be more thorough in evaluating the propagating velocity of UBW pulse of different body types and masses. As the "average male thorax diameter" would not fit everyone.

---

### Official Review · Reviewer_6xgx · 2024-08-11
**Review: Respiro: Continuous Respiratory Rate Monitoring During Motion via Wearable Ultra-Wideband Radar**

**Overall Rating:** 7
**Confidence:** 3

**Other Quality Metrics:**

- Clarity of writing: ***excellent***
- Clinical Significance: ***good***
- Methodological Novelty: ***great***
- Experiments and Results: ***great***

**Questions For The Authors:**

- The demographics of participants seemed to be limited. Would the accuracy of predicted respiratory rate be prone to a more complex data?

**Strengths:**

- Clear and aesthetic visualization is presented in the manuscript for both hardware and software structure and results.
- Data processing method is mathematically supported.
- Comprehensive empirical experiment is conducted with significant improvement over other studies.

**Summary Of The Paper:**

Continuously monitoring respiratory rates outside clinical settings is challenging due to the obtrusive nature and sensitivity to body motions in existing monitoring approaches. This manuscript proposed a signal processing pipeline extracting respiratory signal from windowed channel impulse responses using consumer grade UWB radar modules.

**Weaknesses:**

- Multiple methods on contact/noncontact based RR monitoring are referenced, but only limited methods are implemented as baseline. It would be great to discuss why the rest of the methods are not comparable.

---

### Decision · Program_Chairs · 2024-09-23

Accept